# Paternity Assignment in White Guinea Yam (*Dioscorea Rotundata*) Half-Sib Progenies from Polycross Mating Design Using SNP Markers

**DOI:** 10.3390/plants9040527

**Published:** 2020-04-19

**Authors:** Prince E. Norman, Agre A. Paterne, Agyemang Danquah, Pangirayi B. Tongoona, Eric Y. Danquah, David De Koeyer, Ugochukwu N. Ikeogu, Robert Asiedu, Asrat Asfaw

**Affiliations:** 1Sierra Leone Agricultural Research Institute, Tower Hill, Freetown PMB 1313, Sierra Leone; 2International Institute of Tropical Agriculture, Ibadan PMB 5320, Nigeria; p.agre@cgiar.org (A.A.P.); r.asiedu@cgiar.org (R.A.); a.amele@cgiar.org (A.A.); 3West Africa Centre for Crop Improvement, College of Basic and Applied Sciences, University of Ghana, Legon PMB LG 30, Ghana; adanquah@wacci.ug.edu.gh (A.D.); tongoona@ukzn.ac.za (P.B.T.); edanquah@wacci.ug.edu.gh (E.Y.D.); 4Fredericton Research and Development Centre, Agriculture and Agri-Food Canada, P.O. Box 20280, Fredericton, NB E3B 4Z7, Canada; david.dekoeyer@canada.ca; 5Integrative Plant Breeding, Cornell University, Ithaca, New York, NY 14850, USA

**Keywords:** *Dioscorea*, parentage analysis, pedigree reconstruction, hybrid progeny, DArT markers

## Abstract

White Guinea yam is mostly a dioecious outcrossing crop with male and female flowers produced on distinct plants. Fertile parents produce high fruit set in an open pollination polycross block, which is a cost-effective and convenient way of generating variability in yam breeding. However, the pollen parent of progeny from polycross mating is usually unknown. This study aimed to determine paternity in white Guinea yam half-sib progenies from polycross mating design. A total of 394 half-sib progenies from random open pollination involving nine female and three male parents was genotyped with 6602 SNP markers from DArTSeq platform to recover full pedigree. A higher proportion of expected heterozygosity, allelic richness, and evenness were observed in the half-sib progenies. A complete pedigree was established for all progenies from two families (TDr1685 and TDr1688) with 100% accuracy, while in the remaining families, paternity was assigned successfully only for 56 to 98% of the progenies. Our results indicated unequal paternal contribution under natural open pollination in yam, suggesting unequal pollen migrations or gene flow among the crossing parents. A total of 3.8% of progenies lacking paternal identity due to foreign pollen contamination outside the polycross block was observed. This study established the efficient determination of parental reconstruction and allelic contributions in the white Guinea yam half-sib progenies generated from open pollination polycross using SNP markers. Findings are useful for parental reconstruction, accurate dissection of the genetic effects, and selection in white Guinea yam breeding program utilizing polycross mating design.

## 1. Introduction

Yam belong to the genus *Dioscorea* of the family Dioscoreaceae, order Dioscoreales. *Dioscorea* is the largest genus of Dioscoreaceae comprising about 614 species [1]. Yam species are categorized into different sections. The five most important sections are *Enantiophyllum, Lasiophyton*, *Combilium, Opsophyton* and *Macrogynodium* [2]. *Dioscorea rotundata* Poir, *D. alata* L., *D. cayenensis* Lam, *D. opposite* Thumb and *D. japonica* Thunb belong to *Enantiophyllum* and are distinguished by clockwise twining on support [2]. White Guinea yam (*Dioscorea rotundata* Poir) is native to West Africa [3]. White Guinea yam possesses an allotetraploid structure with disomic inheritance [4,5]. Allotetraploids originate from the merging of two species’ genomes of two sets of homologous chromosomes, with each chromosome pairing only with its homologous form during meiosis [6] to form bivalents [7] and disomic inheritance. This leads to the transmission of heterozygous gametes to the progeny [8].

White Guinea yam is a highly prized African food crop cultivated for its underground starchy tubers. It is a food of choice in West Africa, where it has a deep-rooted connection with the social and cultural facets of the people [9]. White Guinea yam holds tremendous potential as a source of food and income for more than 300 million people [10] and as a raw material for industrial applications. However, the crop’s potential is yet not fully exploited.

Breeding is among the tools to release the yam potential as food source and industrial raw material. Yam breeding utilizes both sexual and asexual reproduction where an artificial hand or natural open pollination applied for the generation of full-sib or half-sib progenies segregating for desired traits and subsequent selection of their clonal derivatives for performance reproducibility and stability over seasons and locations [11]. Yam species often produce male and female flowers on separate plants. Hence, it requires the establishment of separate crossing blocks of male and female parents with multiple planting dates to enhance the synchronization of flowering for hybridization. However, executing planned and controlled crosses are often constrained by the tiny size of flowers, the availability of fertile female and male plants, and the short pollination window due to partial asynchrony [12]. Half-sib breeding, which entails the random open pollination among desirable parents, is a viable alternative to the full-sib breeding which utilizes planned and controlled artificial pollination [13]. Open pollinations involving fertile parents in polycross blocks and trial fields produce a high fruit-set, and hence a cost-effective and convenient way to produce large numbers of seedlings for selection [12].

Open pollination produces half-sib progenies with unstructured pedigree. DNA profiling of progeny and possible parents and comparing their alleles for determination and validation of existing relationships [14] is a viable tool to elucidate the identity of half-sib progenies and reconstruct the pedigree in the outcrossing crops [15]. The technique works on the principle that progeny constitutes allelic contributions from maternal and paternal parents [16]. Such an analysis can provide a reliable estimation of paternal breeding values in the half-sib family and reduces crossing and labeling errors associated with conventional hand pollination. Zoundjihékpon et al. [17] performed the first parentage analysis in cultivated yam, applying six isozyme markers. They validated the progenies of crosses involving well known genitors (one male and three females). Sartie and Asiedu [18] employed nine SSR markers to determine the success of the hybridization of seven *D. rotundata* and *D. alata* mapping populations. The progenies of each of the mapping populations showed combinations of their parental alleles, indicating the success of hybridization.

Paternity testing improves selection gains by increasing parental control in the selection gain equation [19]. Accurate determination of parentage and pedigree relationships helps in determining trait inheritability and ascertaining the genetic progress [20]. Despite the enormous merits of parentage analysis using DNA markers, the technique has not been fully utilized to exploit the potential of open pollination in polycross blocks for yam breeding. The objective of this study was to assess the potential of parentage analysis in determining paternity in the white Guinea yam half-sib progenies generated from open pollination polycross blocks.

## 2. Results

### 2.1. Heterozygosity and Genetic Diversity

The genetic parameter estimates of half-sib progenies are presented in Table 1. The mean proportion of heterozygosity ranged from 0.272–0.328; the minimum proportion of heterozygosity ranged from 0.116–0.294; and the maximum proportion of heterozygosity ranged from 0.360–0.531. The genetic diversity differed among the polycross-derived families. Family TDr1686 had the widest (0.116–0.531) proportion of heterozygosity compared to family TDr1691 (0.216–0.360), which was among families with the narrowest gene diversity. The average distance between the genetic values and the mean (standard deviation) was highest (0.065) for family TDr1686, whereas family TDr1688 exhibited the lowest genetic value deviation from the mean, 0.02. The average minor allele frequency ranged from the highest of 0.238 in family TDr1689 to the lowest, 0.185 in family TDr1690. The average major allele frequency ranged from the highest of 0.813 in family TDr1690 to the lowest, 0.762 in family TDr1689.

### 2.2. Parental Reconstruction and Gametic Composition in Half-Sib Progenies

Generally, there were unequal proportions of half-sib progenies confirmed as true hybrids among crossing parents (Table 2). Of the 394 half-sib progenies sampled, 352 (96.2%) were with fully recovered pollen parent identity, whereas 3.8% lacked paternal identity (Table 2). The pollen parent contribution to progenies varied among male parents utilized in the crosses. The paternal contribution to the progenies was highest for male parent TDr9501932 (65.63%), followed by TDr9902607 (24.43%) and lowest for TDr8902789 (9.94%). Families TDr1685 and TDr1688 had 100% of progenies with fully recovered pedigree, whereas family TDr1689 had the lowest proportion of offspring (56%) with successfully assigned paternity. Of the 50 progenies assessed in family TDr1685; 52, 46 and 2% were contributions from male parents TDr9501932, TDr9902607 and TDr9902789, respectively (Table 2, Figure 1 and Figure 2).

The pedigree recovery of families TDr1686, TDr1687, TDr1688, TDr1689, TDr1690 and TDr1691 are displayed in Figure 2, Figure 3, Figure 4, Figure 5, Figure 6, Figure 7 and Figure 8. Paternal parents TDr9501932 and TDr9902607 contributed 24.0 and 50.0% to family TDr1686; 46.0 and 54.0% to family TDr1688; 86.7 and 11.1% to family TDr1690; 91.8 and 6.1% to family TDr1691 with no contributions from paternal parent TDr9902789 (Table 2). Family TDr1692 had the second-highest proportion (98%) of progenies with fully recovered paternal identity with male parents TDr9501932 and TDr9902789 contributing 44 and 54%, respectively (Table 2 and Appendix A).

We also checked maternal identity in the progenies and found a mismatch in one out of the eight maternal parents that successfully produced offspring from open pollination polycross block. Our analysis revealed that maternal parent ‘Ojuiyawo’ had a low allelic contribution to the progenies despite the highest putative male contributors assigned to the progenies in family TDr1692 (Appendix A).

The proportion of gametes in the studied materials are presented in Table 3. Overall, the eight polycross families exhibited a higher proportion of nonmissing gametes. Family TDr1686 had the highest proportion of nonmissing gametes (94.9%) compared to family TDr1690 (92.6%), which had the lowest. The missing gametes ranged from 5.1 to 7.4% with family TDr1686 (7.4%) exhibiting the highest missing gametes, while family TDr1690 (5.1%) had the lowest.

### 2.3. Numbers and Patterns of SNP Mutations in Half-Sib Progenies of White Yam

Transition SNPs were higher in number and frequency in all progeny families studied (Table 4). The C to T transition accounted for the highest frequency ranging from 30.87 (family TDr1685) to 32.62% (family TDr1687), while the G to C transversion occurred at the lowest frequency ranging from 6.48 (family TDr1691) to 7.47% (family TDr1689). The Tv to Ts ratio ranged from 1:0.60 (family TDr1687) to 1:0.67 (family TDr1685) with a mean of 1:0.64.

## 3. Discussion

### 3.1. Allelic Diversity and Parental Reconstruction

Yam breeding is a two-step process that involves both sexual and asexual reproduction. Sexual reproduction generates recombinant progenies either through artificial hand or natural open pollination. Generating a reasonable number of recombinant progenies for valid selection from potentially interesting parental sources with the controlled cross is a challenging task in yam due to its tiny sized flowers and partial asynchrony [12,13]. However, random open pollination involving fertile parents in polycross blocks and trial fields often produces high fruit-set, and hence, is a cost-effective and convenient way to produce large numbers of seedlings for selection [12,13]. This study assessed the potential of parentage analysis in maximizing heterozygosity in the progenies from an open pollination mating design involving potential parents with a new source of genes. Natural open pollination in white yam generated half-sib families with a variable proportion of heterozygosity. Progenies of half-sib family TDr1686 exhibited the widest (0.116–0.531) heterozygosity and highest standard deviation compared to the remaining half-sib families. Families with higher average distance between the genetic values and the mean (standard deviation) implies that their data points are spread over a large range of values, whereas those with low standard deviation had data points that are close to the mean. The high heterozygosity in some of the families indicated the high genetic variability for a response to within-family selection, whereas the within-family insufficiency in heterozygosity implied the low genetic diversity to respond to within-family selection. Shete et al. [21] also noted the influence of heterozygosity and polymorphic information content on the genetic variability and response to selection. The allelic diversity and genetic differences have been reported to be influenced by the genome transmission mode and number of successful hybridizations [22]. The allelic richness and the initial allelic composition determine the selection and the response to selection, respectively [23,24]. The high allelic richness also indicates higher accessibility of a more substantial fraction of the genotypic space by fewer mutational events [25]. The allelic richness and heterozygosity of a breeding population may be recovered by gene flow induced by migrants carrying lost alleles [26]. Thus, open pollination might facilitate the introduction of new alleles from diverse origins to broaden the genetic base in yam breeding programs.

Sexual fertility and flowering synchrony are essential for success with open natural pollination in yam. Our results showed variations in the proportion of gametes among families (Table 3). Generally, a low proportion of sterile gametes were produced among the parental clones. The observed variation in the proportion of gametes was possibly due to genotype composition, cross-compatibility, and allelic diversity among the parental genotypes. The pedigree reconstitution in studied materials showed that the male and female parents differed widely in terms of their parental ability to produce successful offspring (Table 2). More specifically, a low number of half-sib progenies (only 2–12%) was derived from the pollen parent TDr9902789 compared to the others. This could be attributed to pollen incompatibility, bad pollen shedding, or low pollen viability. Variation in flowering times between the male and female parents could not be the possible cause. Sequential planting (three planting dates in 10 days interval) was employed to facilitate flowering synchronization among the parents. As a result, all the female parents except TDr08-21-3 (Ekpe II) successfully produced progenies. Female parent TDr08-21-3 (Ekpe II) did not flower during the season, which resulted in no fruit and seed sets. Being shy to flower is a common phenomenon in yam [13]. Conversely, the maternal parent ‘Ojuiyawo’ had a low allelic contribution to the progenies despite the highest putative male contributors assigned to the progenies in family TDr1692. The maternal mismatch in family TDr1692 was highly probable to the mishandled maternal identity in field or laboratory during the DNA extraction. Labeling error is the potential cause of mismatch of parents and progenies in the breeding program [27]. This signifies the importance of breeding trial information management for tracking materials using barcodes in the field and during laboratory analysis to avoid errors.

More interestingly, the paternal contribution under natural open pollination was unequal, suggesting unequal pollen migrations or gene flow among the crossing parents. The causes of the varying parental ability and their unequal contribution to progenies have also been noted in other species. In *Hibiscus moscheutos* [28], *Betula pendula* [29] and *Picea abies* [30], unequal paternal contribution was found to influence the pollen-tube growth rate. Spira et al. [31] also observed that the time of arrival of pollen on the stigmatic surface influences paternal contribution in *Hibiscus moscheutos*. Paternal contributions were affected by pollen–pollen interactions in artificial crossing experiments using pollen mixtures in *Pinus sylvestris* [32]. Moreover, paternal contributions were noted to be affected by genetic incompatibility between male and female gametophytes in *Pseudotsuga menziesii* [33] and pollen germination rate in polycrosses of *Cryptomeria japonica* [34]. Detailed studies on these aspects form part of future studies in yam. We also found a low proportion (3.8%) of progenies lacking paternal identity, possibly due to low foreign pollen contamination outside the polycross block. Our polycross block was established in an isolation field with no yam plants in approximately 500 m surroundings. Yam pollen is sticky and the chance for wind pollination is minimal. Flower thrips (*Thysanoptera*) usually play key role with natural open pollination in yam [35]. However, insects belong to *Coleoptera*, *Diptera*, *Hymenoptera* and *Hemiptera* reported visiting the yam field during the flowering period presumably contribute to natural open pollination [36].

The average minor allele frequency of the risk allele tested for all families was greater than 10% indicating its significance in detecting genetic effects in the studied populations. The findings are also in concurrence with Ardlie et al. [37], who noted that loci with high minor allele frequency have a higher power to detect weak genetic effects compared with those with lower minor allele frequency values.

Reliable pedigree information is useful for the accurate dissection of the genetic effects, and thereby maximize selection gain in crop breeding. Molecular markers have been effectively applied in many crops for the reconstruction of pedigree and understand the footprint of breeding history of contemporary germplasms [14,38]. Our result with the application of SNP markers from the DArT genotyping platform successfully recovered paternity in half-sib progenies with known maternity in *D. rotundata*. This suggests the enormous potential of parentage analysis for selection, pedigree identification, and accurate dissection of the genetic effects that could contribute to accurate prediction of the extent of gene flow in heterogeneous populations of yam, as well as other root and tuber crops. Similar analysis could also be applied to other crops too. Telfer et al. [15] reported a high throughput and reproducibility qualities of SNPs in paternity determination and pedigree reconstruction in *Eucalyptus nitens*. Sartie and Asiedu [18] reported the efficiency of molecular markers in confirming the true hybridity of progenies from *D. rotundata* and *D. alata* controlled pairwise crosses. Tamiru et al. [39] confirmed parental line-specific heterozygous SNP markers on 150 F1 individuals obtained from a bi-parental cross in white Guinea yam. They also identified parental-specific region of the genome linked with flower sex determination in white Guinea yam using whole-genome resequencing of bulked DNA samples from female and male F1 progenies, that confirmed that all the progenies were from a specific bi-parental cross. Although genetic variation in breeding populations are well known to be mainly created through mutation, population gene flow, sexual and asexual recombination, the genetic diversity in the current study may have been more attributable to sexual recombination and gene flow.

Parental reconstruction has also been successfully done in other root and tuber crops such as cassava (*Manihot esculenta* Crantz) [40,41,42], potato (*Solanum tuberosum* L.) [43,44] and sweetpotato (*Ipomoea batatas* L.) [45]. In these crops, molecular parentage analysis facilitated the discovery of pedigree errors or mismatches in populations improved using natural and artificial mating schemes [44,46,47]. A detailed review on parentage analysis of root and tuber crops has been reported by Norman et al. [27].

### 3.2. Numbers and Patterns of SNP Mutations in Half-Sib Progenies of White Yam

The numbers and patterns of SNP mutations in this study indicate a bias in chloroplast genome evolution in half-sib progenies of white yam. The diversity and patterns of SNP mutations in the studied half-sib progeny families were possibly due to the function of genes as previously suggested by Cao et al. [48]. The mean Tv to Ts ratio in the eight progeny families was 1:0.64 indicating a bias in favor of transition. This finding is similar with the Tv to Ts ratio of 1:0.60 reported for *D. polystachya* plastid genomes [48]. Small mutations at high frequency have also been reported in plants [49]. However, the frequency of these mutations is suggested to be depleting in coding regions of plant genome compared to those in the noncoding regions [49]. The scope of the present study did not cover comparison of mutational events in the coding and noncoding regions of the white yam genome. This aspect may form part of future genomic research in this plant organism.

## 4. Materials and Methods

### 4.1. Experimental Materials

The experimental materials were botanic seeds generated using polycross mating designs. The polycross design was established at the International Institute of Tropical Agriculture, Ibadan, Nigeria, using 12 genotypes of *D. rotundata* comprising nine females and three males with desired complementary traits for fresh tuber, dry matter, tuber shape, earliness and tolerance to yam mosaic virus [13]. The mating scheme utilized targeted crossing of three female parents to a male parent (3:1) producing nine cross combinations. Staggered planting (three plants at 10 days interval) applied to synchronize flowering among the parents. However, progenies of eight families were successfully generated and utilized in this study except female parent, TDr08-21-3 (Ekpe II) that did not flower to produce viable seed. The botanic seeds generated from the eight families were processed from trilobated fruits harvested from maternal parents in late February 2017.

Prior to establishing the prenursery and nursery, cocopeat and topsoil samples were collected and analyzed at IITA soil analytical lab using standard procedures described by the International Soil Reference and Information Centre (ISRIC) and the FAO [50]. The chemical properties of the cocopeat medium used in the prenursery include 4.61 pH, 0.133% N, 1.401% OC, 0.057% P, 1.618% Ca, 0.199% Mg, 0.380% K, 0.339% Na, 0.008% Zn, 0.002% Cu, 0.019% Mn and 0.006% Fe. The chemical properties of the topsoil medium used in the nursery included 6.08 pH, 0.078% N, 1.489% OC, 0.0004% P, 0.050% Ca, 0.019% Mg, 0.007% K, 0.003% Na, 0.001% Zn, 0.002% Cu, 0.043% Mn and 0.015% Fe.

The prenursery was laid out in a completely randomized design with two replicates in the glasshouse. The cocopeat was put in perforated polyethylene seedling trays and slightly soaked. Different botanic seeds were sown in 5 cm holes created in the growth medium in early March 2017. The trays were well labeled at planting and irrigated every other day until 4 weeks after sowing (WAS). The growing seedlings were fertigated to field capacity every other day for 4 WAS; and fumigated with cypermethrin at the rate of 15 mL per 1 L water at 6 WAS prior to transplanting in the seedling nursery at 8 WAS.

The nursery was laid out in a randomized complete block design in the screen house. Seedlings were transplanted in early June in blocks of 20 plots per block, with inter- and intra-spatial block spaces of 0.5 m apart. The seedlings were transplanted in holes created in the crest of the sterilized topsoil and irrigated to field capacity every three days until 6 months after transplanting (MAT).

### 4.2. Genotyping

Approximately 1 g young, healthy and fully expanded leaves were collected per genotype. Approximately 54 samples comprising 50 progenies, three putative male and one maternal-bearing fruit parents were collected per family. Genomic DNA was extracted from representative samples using modified CTAB protocol with slight modification [51]. The genomic DNA quality and concentration were done using agarose gel and nanodrop as described by Aljanabi and Martinez [52]. Concentrated DNA of 50 µL from each sample was sent to Diversity Array Technology (DArT) Pty Ltd., Canberra, Australia for sequencing.

### 4.3. Analysis

The raw HapMap file from DArTseq genotyping platform (Australia) was first converted to a variant call format (VCF) file using perl programming language and TASSEL v.5.2.43 [53,54]. The output was scored for presence or absence of DNA fragments in genomic sequences generated from genomic DNA samples. A total of 20,000 SNP derived DArTseq markers were identified from the raw data and subjected to filtering using VCFtools [55] at the following conditions: MAF (0.05), no missing data, depth (>5), genotype quality (GQ = 20), maximum and minimum allele = 2, reproducibility = 1 and no indels. Noninformative markers or missing data (in clone and SNP) in the raw data of the VCF file were filtered out. Of the 20000 SNPs subjected to filtering, highly informative 6602 were retained for different analyses. Dosage format was generated for the 6602 informative SNPs using plink and the recodeA function alongside with the double-id [56].

Various population genetic analyses were conducted to explore the genetic properties of the markers. The number of alleles and allele frequencies for the selected SNPs were estimated using the VCFtools [55]. The proportion of heterozygous genotypes also referred to as the gene diversity (GD) of a locus under Hardy-Weinberg equilibrium [57], the minor allele frequency, major allele frequency and the proportions of nonmissing and missing gametes were assessed using TASSEL software version 5.2.51 [53] and the plink command–freq and–hardy [56].

The Penalized and doMC libraries were used to estimate the contribution of the male and female parents in the offspring. The value of each parent was estimated using the multinomial model of the penalized function formula as described by McIlhagga [58].

The log-likelihood of parentage of the observations studied was calculated as:(1)$$L=\overset{n}{\underset{i=1}{\sum }}\overset{q}{\underset{j=1}{\sum }}y_{i,j}\log p_{i,j}.$$
where the (*i*, *j*)th element $y_{i,j}$ is the number of times category *j* occurred in observation *i*. Each element $y_{i,j}$ has a certain probability $p_{i,j}$ of occurrence.
(2)$$P_{j}=\left[\begin{array}{c} \begin{array}{c} p_{1,j}\\ p_{2,j}\\ \vdots \end{array}\\ p_{n,j} \end{array}\right]$$

The observation vectors $y_{1},$
$y_{2},$ ...,${\;y}_{q}$ were stacked into a single vector:(3)$$y=\left[\begin{array}{c} \begin{array}{c} y_{1}\\ y_{2}\\ \vdots \end{array}\\ y_{q} \end{array}\right]$$

The probability vectors $p_{1},$
$p_{2},$ ...,$\;p_{q}$ were used to calculate a single vector:(4)$$p=\left[\begin{array}{c} \begin{array}{c} p_{1}\\ p_{2}\\ \vdots \end{array}\\ p_{q} \end{array}\right]$$

The probability vector ($p_{k}$) of parental contribution to progeny was estimated as:(5)$$p_{k}=\frac{\exp\left(\eta_{k,k}\right)}{\sum_{j=1}^{q}\exp\left(\eta_{k,j}\right)}$$
where each vector $\eta_{k,k}=\;\tilde{X}\beta_{j}$ and the exponentiation and division are both element-by-element; $\tilde{X}$ is a matrix of covariates. The values obtained were then expressed as percentage. The coefficient vectors $\beta_{j}$ was stacked to form a single coefficient matrix.
(6)$$\beta =\left[\begin{array}{c} \begin{array}{c} \beta_{1}\\ \beta_{2}\\ \vdots \end{array}\\ \beta_{q} \end{array}\right]$$

The gradient (intercept) of *L* with respect to *β* was estimated using the chain rule as:(7)$$\frac{dL}{d\beta }=\frac{dp}{d\beta }\;\frac{dL}{dp}$$

The gradient *dL*/*dp* is simply *y*/*p*, where the division is element-by-element. The Jacobian *dp*/*dβ* is
(8)$$\frac{dp}{d\beta }=\left[\begin{array}{ccc} \begin{array}{c} \begin{array}{cc} dp_{1}/d\beta_{1} & dp_{2}/d\beta_{1} \end{array}\\ \begin{array}{cc} dp_{1}/d\beta_{2} & dp_{2}/d\beta_{2} \end{array} \end{array} & \begin{array}{c} \cdots \\ \cdots \end{array} & \begin{array}{c} dp_{q}/d\beta_{1}\\ dp_{q}/d\beta_{2} \end{array}\\ \begin{array}{cc} \vdots & \vdots \end{array} & \ddots & \vdots \\ \begin{array}{cc} dp_{1}/d\beta_{q} & dp_{2}/d\beta_{q} \end{array} & \cdots & dp_{q}/d\beta_{q} \end{array}\right]$$

Each of the submatrices $dp_{k}$/$d\beta_{j}$ is a Jacobian matrix represented as:(9)$$\frac{dp_{k}}{d\beta_{j}}=\frac{dp_{k,1}}{d\beta_{j}}\;\frac{dp_{k}}{d\eta_{k,1}}+\frac{dp_{k,2}}{d\beta_{j}}\;\frac{dp_{k}}{d\eta_{k,2}}+\ldots +\frac{dp_{k,1}}{d\beta_{j}}\;\frac{dp_{k}}{d\eta_{k,q}}=X^{T}\;\frac{dp_{k}}{d\eta_{k,j}}$$
where $dp_{k}$/$d\eta_{k,j}$ is a diagonal matrix of derivatives.

The values generated for each progeny were ranked and the true hybrids were identified based on the estimated male and female coefficients. The profile values of progeny samples of each family and maternal parent were surrounded by the three putative male parents and analyzed for possible alignments. Helium pedigree visualization software was used for pedigree transmission patterns, and visualization of the various polycross families generated as described by Shaw et al. [59] and Shaw [60]. Figure 9 illustrates the process employed for parentage analysis in this paper.

## 5. Conclusions

In the current study, the identification of parentage relationships in the half-sib progenies of white Guinea yam using SNP markers aids the accurate traceability of genotypes with unknown identity, and determination of genetic diversity across families and generations that could be exploited for breeding.

## Figures and Tables

**Figure 1 plants-09-00527-f001:**
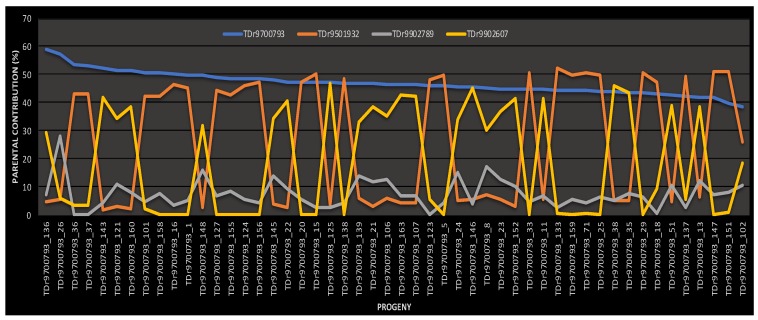
Pedigree reconstruction of polycross family TDr1685 based on genetic contribution of shared parental alleles in progeny. TDr9501932, TDr9902789 and TDr9902607 = male parents; TDr9700793 = female parent.

**Figure 2 plants-09-00527-f002:**
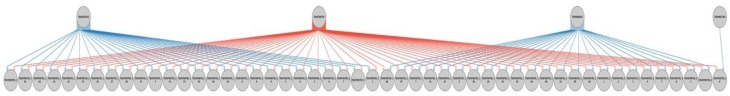
Pedigree reconstruction of polycross family TDr1685 based on Helium pedigree visualization. TDr9501932, TDr9902789 and TDr9902607 = male parents (blue); TDr9700793 = female parent (red).

**Figure 3 plants-09-00527-f003:**
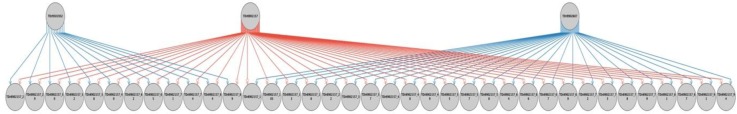
Pedigree reconstruction of polycross family TDr1686 based on Helium pedigree visualization. TDr9501932 and TDr9902607 = male parents (blue); TDr8902157 = female parent (red).

**Figure 4 plants-09-00527-f004:**
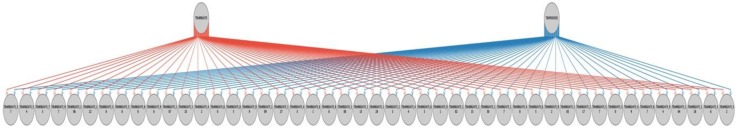
Pedigree reconstruction of polycross family TDr1687 based on Helium pedigree visualization. TDr9501932 = male parent (blue); TDr8902475 = female parent (red).

**Figure 5 plants-09-00527-f005:**
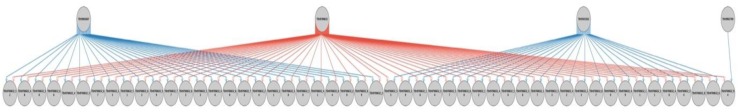
Pedigree reconstruction of polycross family TDr1688 based on Helium pedigree visualization. TDr9902607, TDr9501932 and TDr9902789 = male parents (blue); TDr9700632 = female parent (red).

**Figure 6 plants-09-00527-f006:**
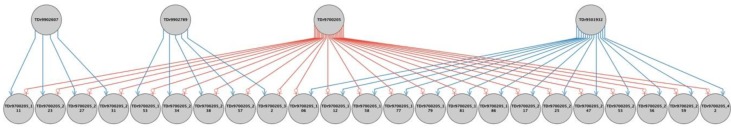
Pedigree reconstruction of polycross family TDr1689 based on Helium pedigree visualization. TDr9501932, TDr9902789 and TDr9902607 = male parents (blue); TDr9700205 = female parent (red).

**Figure 7 plants-09-00527-f007:**
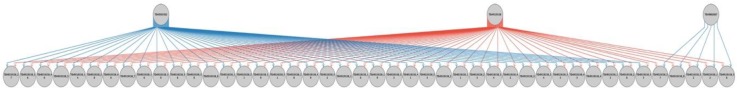
Pedigree reconstruction of polycross family TDr1690 based on Helium pedigree visualization. TDr9501932 and TDr9902607 = male parents (blue); TDr9519158 = female parent (red).

**Figure 8 plants-09-00527-f008:**
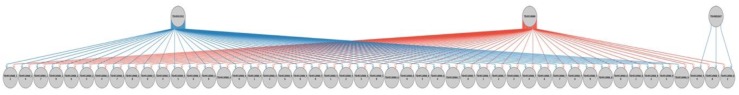
Pedigree reconstruction of polycross family TDr1691 based on Helium pedigree visualization. TDr9501932 and TDr9902607 = male parents (blue); TDr9518988 = female parent (red).

**Figure 9 plants-09-00527-f009:**
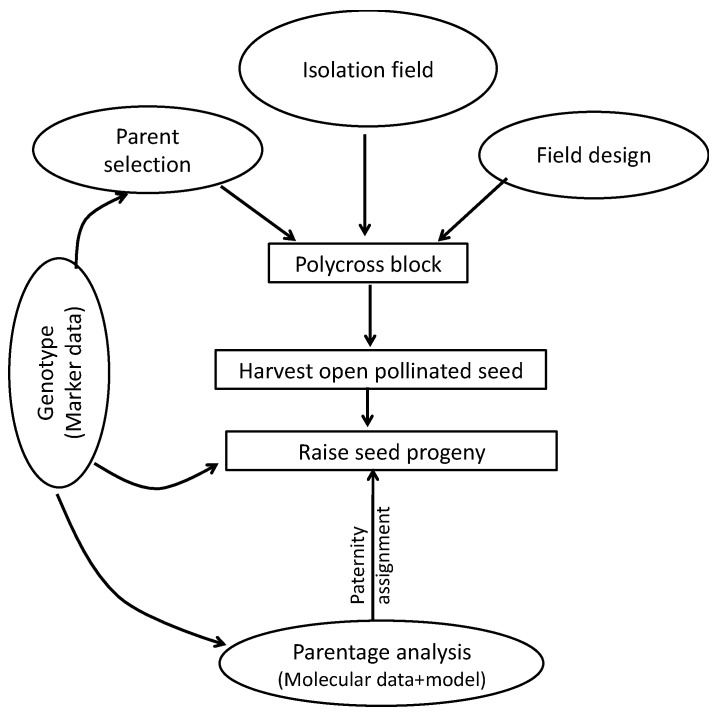
A flow diagram showing the experimental procedure of parentage analysis in white yam.

**Table 1 plants-09-00527-t001:** Summary statistics: minor allele frequency, major allele frequency, proportion of heterozygosity, and QC = quality check of genetic (6602 SNPs, 394 individuals) of segregating progenies of white yam.

Family	Sample	Sample after QC	Hybrid	Proportion of Heterozygous					Minor Allele Frequency	Major Allele Frequency
				Mean	Min	Max	$\sigma^{2}$	σ		
TDr1685	50	50	50	0.276	0.193	0.486	0.002	0.043	0.205	0.794
TDr1686	50	50	37	0.276	0.116	0.531	0.004	0.065	0.211	0.788
TDr1687	50	46	46	0.291	0.229	0.397	0.001	0.030	0.209	0.791
TDr1688	50	50	50	0.327	0.294	0.407	0.000	0.020	0.233	0.766
TDr1689	50	28	28	0.328	0.161	0.517	0.002	0.048	0.238	0.762
TDr1690	45	44	44	0.272	0.255	0.437	0.001	0.029	0.186	0.813
TDr1691	49	48	48	0.281	0.216	0.360	0.001	0.029	0.204	0.795
TDr1692	50	50	49	0.293	0.251	0.460	0.001	0.038	0.209	0.791

$\sigma^{2}$ = variance, and σ = standard deviation.

**Table 2 plants-09-00527-t002:** The number and proportions of progenies with recovered paternal identity.

Female	Family	Male			Progeny with Fully Recovered Pedigree	Percent of Progenies with Fully Recovered Pedigree
		TDr9501932	TDr9902789	TDr9902607		
TDr9700793	TDr1685	26 (52%)	1 (2.0%)	23 (46%)	50	100
TDr8902157	TDr1686	12 (24%)	0 (0%)	25 (50%)	37	74
TDr8902475	TDr1687	46 (92%)	0 (0%)	0 (0%)	46	92
TDr9700632	TDr1688	23 (46%)	1 (2%)	26 (52%)	50	100
TDr9700205	TDr1689	18 (36%)	6 (12%)	4 (8%)	28	56
TDr9519158	TDr1690	39 (86.7%)	0 (0%)	5 (11.1%)	44	97.8
TDr9518988	TDr1691	45 (91.8%)	0 (0%)	3 (6.1%)	48	97.9
Ojuiyawo	TDr1692	22 (44.0%)	27 (54%)	0 (0%)	49	98
Total progenies		231	35	86	352	
% contribution		65.63	9.94	24.43	100	96.2

**Table 3 plants-09-00527-t003:** Gamete frequency of eight polycross-derived families of yam.

Family	Number of Nonmissing Gametes	Proportion of Nonmissing Gametes	Number of Missing Gametes	Proportion of Missing Gametes
TDr1685	669930	0.940	42870	0.060
TDr1686	660336	0.926	52680	0.074
TDr1687	672606	0.943	40410	0.057
TDr1688	676128	0.948	36888	0.052
TDr1689	681738	0.939	44482	0.061
TDr1690	601600	0.949	32192	0.051
TDr1691	643928	0.938	42680	0.062
TDr1692	674222	0.946	38794	0.054

**Table 4 plants-09-00527-t004:** Percentage of transition and transversion SNPs across the *Dioscorea rotundata* genome.

Family	SNP Type						Tv to Ts Ratio
	Transitions (Ts)		Transversions (Tv)				
	AG	CT	AT	AC	GT	GC	
TDr1685	805 (29.10)	854 (30.87)	394 (14.24)	257 (9.29)	261 (9.44)	195 (7.05)	1.67
TDr1686	822 (28.20)	938 (32.18)	413 (14.17)	251 (8.61)	280 (9.61)	211 (7.24)	1.66
TDr1687	785 (30.05)	852 (32.62)	344 (13.17)	227 (8.69)	230 (8.81)	174 (6.66)	1.60
TDr1688	829 (28.57)	924 (31.84)	416 (14.33)	259 (8.92)	266 (9.17)	208 (7.17)	1.66
TDr1689	869 (27.86)	1005 (32.22)	419 (13.43)	292 (9.36)	301 (9.65)	233 (7.47)	1.66
TDr1690	756 (29.62)	820 (32.13)	349 (13.68)	223 (8.74)	228 (8.93)	176 (6.90)	1.62
TDr1691	755 (29.13)	841 (32.45)	353 (13.62)	237 (9.14)	238 (9.18)	168 (6.48)	1.62
TDr1692	831 (29.37)	888 (31.39)	403 (14.25)	256 (9.05)	256 (9.05)	195 (6.89)	1.65
Mean	806.5 (28.99)	890.25 (31.96)	386.38 (13.86)	250.25 (8.98)	257.5 (9.23)	195.0 (6.98)	1.64

Values in brackets are percentages.

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
