# Peer review of "Paternity Assignment in White Guinea Yam (Dioscorea Rotundata) Half-Sib Progenies from Polycross Mating Design Using SNP Markers"

_plants, 2020, doi:10.3390/plants9040527_

Round 1
Reviewer 1 Report
Title: Paternity assignment in white Guinea yam 2 (Dioscorea rotundata) half-sib progenies from 3 polycross mating design using SNP markers
Summary
Authors were interested to determine paternity in white Guinea yam half-sib progenies from polycross mating design. They used a total of 394 half-sib progenies from random open pollination involving nine female and three male parents was genotyped with 6602 SNP markers from DArTSeq platform to recover full pedigree. They showed efficient determination of parental reconstruction and allelic contributions in the white Guinea yam half-sib progenies generated from open pollination polycross using SNP markers.
The study is interesting but l have suggestions to improve further.
Major points
- Please represent the entire process in a flow diagram. It will help readers to understand better.
- Table 1: Please include variance and standard deviations to make the table comprehensive.
- Figure 1(a),1(b) and figure 2 needs really major attention. Please represent the same information in different way. It’s very confusing what message are coming out. Also figure quality needs to improve significantly.
- Line 283 : the log likelihood came from which equation? What model was fitted
- Line 297: dL/dB os not clear
- Can similar approach can be applied in the some other crops? Please discuss this in the discussions.
Minor points
- Supplementary figure quality needs to improve significantly.
Author Response
Dear Sir,
On behalf of colleague authors, I write to thank the reviewer for his inputs and provide answers to questions raised on our manuscripts.
All revisions effected based on the reviewers suggstions are highlighted red for easy tracking.
The flow diagram has been included in the revised manuscript.
The variance and standard deviation have been in table 1, reported and discussed as suggested.
The multilinear model used and equations have been included and well explained.
Some crops in which parentge analysis has been done have been included and discussed.
Best regards,
P.E. Norman

Reviewer 2 Report
DArTSeq markers have been successfully utilized in reconstructing the pedigrees in white Guinea yam. The outcome of this research will be of immense help in improving yam yields and stress tolerance. Experimental design and data analysis are very clearly described. Overall, the manuscript is very interesting, well written, and the findings will aid in marker-assisted yam breeding to provide food security in the African continent.
There are a few suggestions/concerns that needed to be addressed in the manuscript.
Major suggestions:
Table 2: Why were low numbers of half-sib progenies (only 2-12%) derived from pollen parent TDr9902789? Is it due to pollen incompatibility, bad pollen shedding, or due to huge variation in flowering times between TDr9902789 and the female parents? Please include the rationale in the discussion, as it is not clearly mentioned in the present version.
What is the recommended isolation distance between mating blocks for yam to prevent foreign pollen contamination? How far were foreign yam pollen parents to this mating block?
Figures 1B, 2, 3, and S1–S6. In Helium pedigree visualization, parental genotypes are not clearly visible. Including a different color for each parent and increasing font size will help with the visualization.
Lines 140–142: Maternal parent ‘TDrOjuiyawo’ had a low allelic contribution to the progenies (Table S1). Include the possible reasons for this issue in the discussion.
Are there any dominant traits in the pollen parents? Were these traits used to validate/correlate with the marker-assisted pedigree reconstruction of half-sib progenies in that half-sib family?
Minor suggestions:
Line 43: Dioscorea rotundata, Poir: Remove comma after the species name
Lines 162–168: This information was already presented in the Introduction and is redundant.
Author Response
Dear Sir,
I write to submit responses to suggestions and questions on our manuscripts. These can be traced in red in the revised version of attached manuscript.
Low number of half sib progenies is partly due to low pollen viability, varying flowering window (Norman et al., 2018).
The recommended isolation distance is 700 m or more.
The Helium pedigree visualization software does not have colour options. As such, black was used.
Other edits were done as suggested except that the quality improvement was unsuccessful.
Best regards,
P.E. Norman

Reviewer 3 Report
This paper is interesting and deserved for publication. However, some places need revision before proceeding further.
- English should be checked and revised. For instance, lines 32-34, "this study" was repeated two times. Similar redundancies and revision need to check throughout the paper. For instance, the line 32, "we also observed that...", should be " a total 3.8% progenies....was observed".
- Lines 89-108, your Figures or Table should be placed under the position of your text.
- Line 236, provide the location.
- Lines 280-282, modify the font type similar with that of the main text, not Time News Roman.
- Supplementary Materials : these down parts should be modified the font type and size.
- Line 128, Table S1, add Supplementary Material Table S1, so the readers can understand better.
This paper has a low rate of similarity, thus its originality is good.
Author Response
Dear Sir,
On behalf of colleague authors, I write to express our thanks for you inputs and respond to concerns raised on our manuscript. All corrections including those from colleague reviewers are highlighted in red.
The experiment was done at the International Institute of Tropical Agriculture (IITA), Ibadan, Nigeria. Other suggestions by you were addressed.
Kind regards,
P.E. Norman

Round 2
Reviewer 1 Report
Authors are take into account all my previous comments
Author Response
Attached herein is the revised document based on the editor's suggestion. The revised portion is highlighted red. Thanking you very much.